# Fe–Ce Bimetallic MOFs for Water Environment Remediation: Efficient Removal of Fluoride and Phosphate

**DOI:** 10.3390/nano15211623

**Published:** 2025-10-24

**Authors:** Jinyun Zhao, Yuhuan Su, Jiangyan Song, Ruilai Liu, Fangfang Wu, Jing Xu, Tao Xu, Jilin Mu, Hao Lin, Jiapeng Hu

**Affiliations:** 1Fujian Provincial Bamboo Engineering Technology Research Center, Wuyishan 354300, China; biaoduo@163.com (J.Z.); suyuhuan0001@163.com (Y.S.); wyulrl@163.com (R.L.); wuff2018@wuyiu.edu.cn (F.W.); Xulao@163.com (T.X.); mujilin@163.com (J.M.); 2College of Environmental Science and Engineering, Tongji University, Shanghai 200092, China; sjya@tongji.edu.cn; 3College of Environment and Safety Engineering, Fuzhou University, Fuzhou 350001, China

**Keywords:** fluoride removal, phosphate adsorption, Fe-Ce-MOFs, hydrothermal synthesis, wastewater purification

## Abstract

Fe–Ce-MOFs with a rice-grain-like morphology were successfully obtained via hydrothermal synthesis, where ferric chloride (FeCl_3_) and cerium nitrate [Ce(NO_3_)_3_] served as the metal precursors and terephthalic acid (PTA) acted as the organic coordinating ligand. The effects of the Fe:Ce molar ratio, (Fe/Ce):PTA ratio, reaction duration, and synthesis temperature on adsorption performance of the Fe–Ce-MOFs were systematically studied. A comprehensive evaluation was conducted on the removal of fluoride and phosphate ions from aqueous solution. Under optimized conditions, the maximum adsorption capacities of Fe–Ce-MOFs for fluoride and phosphate reached 183.82 mg g^−1^ and 110.74 mg g^−1^, respectively. Adsorption data correlated strongly with the Langmuir isotherm, were best represented by the pseudo-second-order kinetic model, and were identified as a spontaneous and endothermic reaction. After three regeneration cycles, the adsorbent still maintained high removal efficiencies for fluoride (85.17%) and phosphate (47.34%) removal. In practical wastewater treatment, removal efficiencies of 92.04% for fluoride and 93.87% for phosphate were achieved. Mechanistic studies revealed that fluoride removal was dominated by electrostatic attraction and hydroxyl–fluoride ion exchange, whereas phosphate removal was attributed to the generation of inner-sphere complexes involving PO_4_^3−^ and Fe/Ce active sites. This study not only elucidates the synergistic mechanism of fluoride and phosphate elimination by Fe–Ce-MOFs but also provides theoretical guidance and application prospects for the development of highly efficient and stable bimetallic MOF-based adsorbents for environmental remediation.

## 1. Introduction

Owing to the accelerated progress of human society, large-scale industrialization and urbanization have significantly aggravated environmental pollution. Fluoride and phosphate, as typical inorganic anion pollutants, are widely present in effluents from metallurgy, glass manufacturing, fertilizer production, electronics, and livestock wastewater. Excessive fluoride in aquatic environments can cause significant human well-being concerns such as skeletal fluorosis and dental fluorosis, while the discharge of phosphate is the main driver of water eutrophication and algal blooms, further threatening ecosystem stability and the safety of potable water. Therefore, high-efficiency removal of fluoride and phosphate from wastewater represents an urgent challenge in environmental science and water treatment [1,2,3].

Currently, common technologies for defluoridation and dephosphorization include precipitation, membrane separation technologies, phytoremediation, and adsorption strategies [4,5]. Among them, the precipitation method is simple and fast but tends to generate large amounts of sludge, leading to secondary pollution. Membrane separation exhibits an excellent removal performance but suffers from high equipment costs and membrane fouling. Ion exchange possesses strong selectivity, yet resin regeneration is complex and costly. Conversely, owing to its facile operational procedure, adsorption is increasingly regarded as a viable and emerging technology, with a fast adsorption rate, effective removal of low-concentration pollutants, and good regeneration potential. Metal–organic frameworks are a unique class of crystalline porous compounds constructed by coordination bonds established by the coordination of metal nodes with organic linkers. MOFs feature remarkable specific surface areas, customizable pore architectures, and abundant functional sites, and they have shown wide application potential in gas separation, catalytic reactions, energy storage systems, and water purification [6,7]. For the removal of fluoride and phosphate, MOFs offer abundant active binding sites attributable to their extensive surface area and multiple coordination centers. However, the limited stability of some MOFs in aqueous environments, together with insufficient adsorption capacity and selectivity, restricts their practical applications [8,9]. Thus, designing MOF-based composites with improved stability and an enhanced adsorption capacity has become a research hotspot.

Previous studies have demonstrated that rare earth metal-based MOFs and transition metal-based MOFs exhibit an excellent performance in fluoride and phosphate removal. Song et al. [10] fabricated rod-like Ce-MOFs via a solvothermal method, achieving the highest fluoride uptake capacity of 129.7 mg g^−1^. The fluoride removal efficiency remained at 72% after five adsorption–desorption cycles, indicating good recyclability. In recent years, iron-based metal–organic frameworks (Fe-MOFs) have received extensive attention in adsorption and water purification due to their abundant active sites, low toxicity, and tunable coordination environment. Numerous studies have demonstrated that Fe-MOFs such as MIL-101(Fe), MIL-53(Fe), and Fe-BTC exhibit an excellent affinity toward various contaminants, including fluoride, phosphate, and heavy metals [11]. Mohammadi et al. [12] fabricated a structurally stable Fe-MOF via an easy single-step solvothermal synthesis, achieving removal efficiencies of 67%, 88%, and 6% for fluoride ions, anthraquinone-based dye (ARS), and triphenylmethane dye (MG), respectively. To enhance the adsorption efficiency of single-metal MOFs, researchers have developed bimetallic or multimetallic MOFs by introducing bimetallic or multimetallic components. Compared with single-metal MOFs, bimetallic MOFs provide multiple active sites through the synergistic effect of different metals, thereby enhancing the affinity for various anions, improving structural stability, and promoting electron transfer. Moreover, bimetallic MOFs usually possess larger surface areas and more open metal sites, leading to superior adsorption capacities and recyclability. For instance, Hu et al. [13] prepared Ce-La-MOFs and systematically investigated their defluoridation performance. An ultimate fluoride uptake capacity of 138.64 mg g^−1^ was observed, and even after four regeneration cycles, the adsorption capacity was sustained at 36.25 mg g^−1^. Similarly, Wang et al. [14] synthesized an Fe-La composite adsorbent possessing a BET surface area of 113.13 m^2^ g^−1^, via a one-pot hydrothermal process. This composite exhibited adsorption capacities of 27.41 mg g^−1^ (fluoride) and 89.41 mg g^−1^ (phosphate), which were achieved within the pH window of 3.8–7.1. These findings suggest that introducing Ce and Fe into bimetallic MOFs not only exploits the strong affinity of Ce for fluoride but also takes advantage of the high phosphate adsorption activity of Fe, providing a feasible pathway for designing new materials with dual defluoridation and phosphate removal functions.

In this work, FeCl_3_ and Ce(NO_3_)_3_ were selected as metal precursors and terephthalic acid (PTA) as the organic ligand to synthesize Fe-Ce-MOFs via a hydrothermal method. The impacts of varying of Fe:Ce molar ratios, the (Fe/Ce):PTA molar ratio, the reaction duration, and the synthesis temperature on the adsorption performance of the resulting Fe-Ce-MOFs were systematically investigated. The adsorption equilibrium data were analyzed using the Langmuir and Freundlich models, while the kinetic behavior was interpreted through pseudo-first-order, pseudo-second-order, and intraparticle diffusion models. In addition, regeneration and reusability tests as well as real wastewater treatment tests were performed to comprehensively determine the practical applicability and removal mechanisms of the Fe-Ce-MOFs.

## 2. Materials and Methods

### 2.1. Chemical Reagents

The reagents used in this study included ferric chloride (analytical grade, Sinopharm Chemical Reagent Co., Ltd., Shanghai, China), cerium nitrate (analytical grade, Shanghai Macklin Biochemical Technology Co., Ltd., Shanghai, China), terephthalic acid (98%, Shanghai Macklin Biochemical Technology Co., Ltd.), hydrochloric acid (36–38%, Sinopharm Chemical Reagent Co., Ltd.), N,N-dimethylformamide (DMF, analytical grade, Sinopharm Chemical Reagent Co., Ltd.), sodium hydroxide, absolute ethanol, and formic acid (analytical grade, Sinopharm Chemical Reagent Co., Ltd.). All other reagents were of analytical grade and used as received without further purification.

### 2.2. Synthesis of Fe-Ce-MOFs

The synthesis began with dissolving 0.12 g FeCl_3_, 0.96 g Ce(NO_3_)_3_, and 0.125 g PTA in 30 mL DMF, assisted by 30 min of ultrasonic irradiation. The solution was moved into a PTFE-lined autoclave and heated at 150 °C for 18 h. Once the system reached room temperature, the solid product was collected, washed 2–3 times with DMF, centrifuged, and vacuum-dried at 50 °C for 12 h. The obtained material was designated as Fe-Ce-MOFs.

To evaluate the effect of the metal molar ratio and the total metal-to-ligand (PTA) molar ratio on the adsorption performance, the following experiments were designed. The total mass of FeCl_3_ and Ce(NO_3_)_3_ was fixed at 1.08 g, and the mass of PTA was set to 0.125 g. Different Fe/Ce molar ratios (1:0.5, 1:1, 1:2, 1:3, 1:4, 1:5) were tested following the same synthesis procedure, and the adsorption performance of the resulting Fe-Ce-MOFs toward fluoride and phosphate was evaluated to determine the optimal ratio. Based on adsorption results, the molar proportion of Fe to Ce (1:3) was selected as the optimal composition. Subsequently, FeCl_3_ and Ce(NO_3_)_3_ were used at this ratio, and different total metal-to-PTA molar ratios (1:2, 1:3, 1:4, 1:6, and 1:8) were investigated by repeating the synthesis procedure. The adsorption performance of the resulting Fe-Ce-MOFs was compared to determine the optimal metal-to-ligand ratio.

### 2.3. Adsorption Experiments of Fluoride and Phosphate

A total of 0.01 g of Fe-Ce-MOF adsorbent was added to 50 mL solutions containing initial fluoride or phosphate concentrations of 20, 25, 30, 35, 40, 45, and 50 mg L^−1^, respectively. The mixtures were shaken for 12 h and then filtered through a microporous membrane.

The fluoride concentration was determined as follows: 10 mL of the supernatant was transferred into a 50 mL volumetric flask, followed by the addition of 10 mL buffer solution and dilution to the mark with deionized water. The fluoride concentration was then measured using the potentiometric method.

The phosphate concentration was determined as follows: a certain volume of filtrate was mixed with 25 mL deionized water, 2 mL ammonium molybdate, and 3 mL ascorbic acid in a 50 mL volumetric flask. The solution was diluted to the mark with deionized water, then incubated in a water bath at 30 °C for 30 min, and the concentration was determined at 880 nm using a UV–Vis spectrophotometer.

The adsorption capacity was calculated using Equation (1), and the removal efficiency was calculated using Equation (2):(1)$$q_{e}=\frac{v(C_{0}-C_{e})}{m}$$(2)$$\eta \%=\frac{C_{0}-C_{e}}{C_{0}}\times 100\%$$
where *q_e_* (mg g^−1^): the equilibrium adsorption capacity; *V* (L): the volume of the initial solution; *C_0_* and *C_e_* (mg L^−1^): the initial and equilibrium concentrations, respectively; *m* (g): the mass of the adsorbent dosage; *η* (%): removal efficiency.

This study also investigated the effect of different pH conditions on the removal efficiency of fluoride and phosphate by the Fe-Ce-MOF adsorbent. The solution pH was adjusted using 0.1 mol L^−1^ NaOH or HCl to evaluate the influence of pH on the stability and adsorption performance of the adsorbent. In the batch equilibrium experiments, the effects of coexisting anions, including Cl^−^, NO_3_^−^, SO_4_^2−^, CO_3_^2−^, HCO_3_^−^, PO_4_^3−^, and F^−^, in the concentration range of 10–100 mg L^−1^ on the removal of fluoride and phosphate were also examined. For the adsorption thermodynamics study, the adsorption efficiency of Fe-Ce-MOFs for fluoride and phosphate was evaluated at 25, 35, and 45 °C. Unless otherwise specified, the experimental conditions were as follows: adsorbent dosage of 0.01 g, initial fluoride and phosphate concentration of 20 mg L^−1^, solution pH of 4, and adsorption time of 12 h. Regeneration experiments of Fe-Ce-MOF adsorbent: The saturated Fe-Ce-MOF adsorbent after adsorption was collected and transferred into a 50 mL centrifuge tube, followed by the addition of 30 mL of 0.1 mol L^−1^ NaOH solution. The mixture was shaken at 150 rpm for 2 h in a thermostatic shaker to achieve the desorption of fluoride or phosphate ions. After desorption, the adsorbent was separated by centrifugation and the supernatant was discarded. The solid was washed three times with deionized water until the washing solution reached a nearly neutral pH. The washed adsorbent was then dried in a vacuum oven at 50 °C for 12 h to obtain the regenerated Fe-Ce-MOFs. The above procedure was defined as one regeneration cycle, and the process was repeated for subsequent cycles (i.e., two times, three times, and so on).

In the treatment of real fluoride-containing wastewater, the field wastewater was collected from Shaowu Industrial Park, Fujian Province, China, and originated from equipment cleaning processes. For the treatment of real phosphorus-containing wastewater, the field wastewater was obtained from effluent generated during equipment cleaning at a fertilizer factory in Shunchang, Fujian Province, China.

### 2.4. Characterization

The morphology and composition of adsorbent were determined by scanning electron microscopy hyphenated with energy dispersive X-ray spectroscopy (SEM-EDX) using equipment from Hitachi High-Tech (Shanghai) Co., Ltd., Shanghai, China. The powder X-ray diffraction (XRD) pattern was recorded using a Bruker D8 ADVANCE with CuK_α_ to examine the crystal structure of the adsorbent, where the specific angle was selected from 5 to 70° and the scanning speed was selected at 10°/min. N_2_ sorption isotherms were measured at −196 °C using a Micromeritics Instrument (ASAP 2420, Micromeritics (Shanghai) Instrument Co., Ltd., Shanghai, China), and the specific surface area was estimated by the Brunauer–Emmett–Teller (BET) method. The thermal stability of the adsorbent was studied by a thermogravimetric analysis (TGA) instrument (DTA-60H, DTG, SHIMADZU, Kyoto City, Japan), in the temperature range from 20 to 800 °C under N_2_ flow. The surface property of the adsorbent was examined using FTIR (Perkin Elmer, Shanghai, China) in the range of 400–4000 cm^−1^ using the KBr disk method. The elemental compositions and chemical valences were determined by XPS (ESCALAB250, Thermo VG, Waltham, MA, USA).

### 2.5. Adsorption Thermodynamics

The experimental data were fitted using the Langmuir and Freundlich isotherm models, expressed as follows:

Langmuir model:(3)$$\frac{C_{e}}{q_{e}}=\frac{1}{K_{L}q_{m}}+\frac{C_{e}}{q_{m}}$$

Freundlich model:(4)$$\mathrm{lg}q_{e}=\mathrm{lg}K_{f}+\frac{1}{n}\mathrm{lg}C_{e}$$
where *q_e_* (mg g^−1^) is the equilibrium adsorption capacity, *q_m_* (mg g^−1^) is the maximum adsorption capacity, *C_e_* (mg L^−1^) is the equilibrium concentration, *K_L_* (L mg^−1^) is the Langmuir constant, *K_f_* (mg g^−1^) is the Freundlich adsorption constant, and n represents adsorption intensity.

To further clarify the adsorption mechanism, thermodynamic parameters including Gibbs free energy change (ΔG^0^), enthalpy change (ΔH^0^), and entropy change (ΔS^0^) were calculated using the following Equations (5)–(7):(5)$$K_{D}=\frac{q_{e}}{C_{e}}$$(6)$$\Delta G^{0}=-RT\ln K_{D}$$(7)$$\ln K_{D}=\frac{\Delta S^{0}}{R}-\frac{\Delta H^{0}}{RT}$$
where *K_D_* is the distribution coefficient, *R* is the universal gas constant (8.314 J mol^−1^ K^−1^), and *T* is the absolute temperature (K).

### 2.6. Adsorption Kinetics

Adsorption kinetics were analyzed by fitting the experimental data with pseudo-first-order, pseudo-second-order, and intraparticle diffusion models to elucidate the adsorption mechanism. Adsorption kinetics mainly describe the dynamic behavior of adsorbates on the adsorbent surface during the adsorption process, including the adsorption rate, influencing factors, and the underlying mechanisms. In this study, the kinetic data were fitted using the pseudo-first-order, pseudo-second-order, and intraparticle diffusion models. The corresponding equations are expressed as follows.

Pseudo-first-order model:(8)$$\ln(q_{e}-q_{t})=\ln q_{e}-k_{1}t$$

Pseudo-second-order model:(9)$$\frac{t}{q_{t}}=\frac{1}{k_{2}q_{e}^{2}}+\frac{t}{q_{e}}$$

Intraparticle diffusion model:(10)$$q_{t}=k_{p}t^{0.5}+C$$
where *q_t_
*(mg g^−1^) is the adsorption capacity at time *t* (min), *q_e_* (mg g^−1^) is the equilibrium adsorption capacity, *k_1_* (min^−1^) is the pseudo-first-order rate constant, *k_2_* (g mg^−1^ min^−1^) is the pseudo-second-order rate constant, *k_p_* (mg g^−1^ min^−1^/^2^) is the intraparticle diffusion rate constant, and *C* is a constant.

## 3. Results and Discussion

### 3.1. Influence of Reaction Parameters on Defluoridation and Dephosphorization

Figure 1a shows the SEM image of the Fe-Ce-MOF adsorbent. The material exhibits a uniformly distributed rice-grain-like morphology with an average particle size of 198 ± 34 nm. EDS mapping confirmed the homogeneous distribution of C, O, N, Fe, and Ce elements without obvious agglomeration, indicating that Ce and Fe were successfully incorporated into the MOF framework. There are slight differences in local details between Figure 1a and the elemental mapping images, mainly because the magnification and detection depth of the EDS mapping differ from those of the SEM image; the EDS maps only display a localized region on the sample surface. Elemental analysis further revealed that the atomic ratio of Fe to Ce was approximately 2:1, consistent with the initial feed ratio. The adsorption experimental procedure and the meaning of each parameter are described in Appendix A. The effect of the Fe:Ce molar ratio on fluoride and phosphate removal is presented in Figure 2a. At an Fe:Ce ratio of 1:0.5, the removal efficiencies of fluoride and phosphate were 80.05% and 54.62%, respectively. With an increasing Ce content, both defluoridation and dephosphorization efficiencies improved, reaching maximum values of 92.93% and 68.40% at an Fe:Ce ratio of 1:3. However, further increases in Ce content led to a decline in removal performance. Based on this result, the Fe:Ce ratio of 1:3 was selected for further optimization. Figure 2b illustrates the effect of the total metal-to-ligand (Fe/Ce:PTA) molar ratio on the adsorption performance. The optimum removal efficiencies for both fluoride and phosphate were achieved at a metal-to-ligand ratio of 1:4. An insufficient PTA content could result in incomplete coordination to metal centers and the generation of crystalline particles, thereby reducing the adsorption efficiency. Conversely, excessive PTA promoted the generation of filamentous species lacking metal coordination, which also hindered the adsorption performance [13].The influence of the reaction duration on fluoride and phosphate removal is shown in Figure 2c. The removal efficiencies increased with an extended reaction time, reaching maximum values at 18 h. Prolonging the reaction time beyond 18 h did not yield a further improvement, and thus 18 h was recognized as the best reaction duration. As shown in Figure 2d, the reaction temperature (120 °C, 150 °C, and 180 °C) had no significant effect on the removal performance. Therefore, 120 °C was selected as the synthesis temperature considering energy efficiency.

### 3.2. Characterization and Analysis of Fe-Ce-MOFs

Figure 3a presents the XRD patterns of Fe-Ce-MOF pre- and post-adsorption. The diffraction peaks at 2*θ* = 16.51°, 23.65°, 28.44°, and 33.02° correspond to the characteristic peaks of Ce-MOF [10,15], while those at 2*θ* = 9.21°, 14.75°, and 41.65° are attributed to Fe-MOF [16,17], confirming the successful synthesis of Fe-Ce-MOFs. After fluoride adsorption, the diffraction peaks weakened or even disappeared, and new peaks appeared at 2*θ* = 20.11° and 31.52°, which were assigned to CeF_3_ [18,19]. This phenomenon indicates that fluoride ions interacted with the exposed Ce^3+^ sites through ion exchange and surface complexation, leading to partial structural collapse of the original MOF lattice and formation of Ce–F bonds. The disappearance of several characteristic MOF peaks suggests that the adsorption process involved both physical capture and chemical reaction mechanisms, resulting in a change in crystallinity. Similarly, after phosphate adsorption, the diffraction peaks also weakened, with new peaks emerging at 2θ = 25.16° and 44.23°, which can be attributed to iron or cerium phosphates formed during the adsorption process [20,21]. These results confirm that both Fe and Ce active centers participated in the adsorption via ligand exchange and electrostatic attraction mechanisms, implying that the defluoridation and dephosphorization are governed by a combination of electrostatic attraction and chemical bonding interactions between metal ions and anions.

The FT-IR spectra of Fe-Ce-MOFs pre- and post-adsorption are shown in Figure 3b. Before adsorption, the peaks at 3482 cm^−1^ and 3129 cm^−1^ are ascribed to the –OH stretching vibrations of surface-adsorbed water [22,23]. Peaks at 1583 cm^−1^ and 1396 cm^−1^ are assigned to the asymmetric and symmetric stretching vibrations of –COOH groups, while those at 1542 cm^−1^ and 1511 cm^−1^ are ascribed to stretching vibration of aromatic C=C bonds [24]. Peaks at 774 cm^−1^ and 753 cm^−1^ are characteristic of M–O coordination bonds [22]. After fluoride and phosphate adsorption, the overall peak positions and intensities remained nearly unchanged, suggesting good structural stability of Fe-Ce-MOFs. However, in the case of phosphate adsorption, new peaks appeared at 1049, 614, and 541 cm^−1^. The band at 1049 cm^−1^ was assigned to asymmetric stretching vibrations of P–O in PO_4_^3−^ [25], while those at 614 and 541 cm^−1^ correspond to O–P–O bending vibrations [26].

The N_2_ adsorption–desorption isotherm of Fe-Ce-MOFs is displayed in Figure 3c. The curve exhibits a typical type IV isotherm with a pronounced increase in adsorption at relative pressures (P/P_0_) of 0.8–1.0, accompanied by adsorption–desorption hysteresis, indicating a mesoporous structure. The distribution of pore sizes (inset) further confirms an average pore diameter of 10.27 nm. BET analysis revealed a specific surface area of 15.95 m^2^ g^−1^ and a pore volume of 0.039 cm^3^ g^−1^. Although the surface area is relatively low, the mesoporous structure provides accessible adsorption sites. The hysteresis loop suggests a “bottle-neck to cavity” type of pore structure, which is commonly observed in multimetallic MOFs, attributable to the synergistic coordination of Fe and Ce centers [27,28].

Thermal stability was investigated using TG–DTG–DSC analysis carried out under N_2_ with a heating rate of 10 °C min^−1^ over the temperature range of 25–800 °C (Figure 3d). The thermal decomposition proceeded in three stages. The first stage (25–250 °C) showed a weight loss of ~3.33%, corresponding to the desorption of physically adsorbed water and residual solvents such as DMF. The second stage (250–335 °C) exhibited ~24.18% weight loss with significant exothermic signals, attributed to the initial decomposition of terephthalic acid ligands. The third stage (369–404 °C) showed the most significant weight loss (~43.76%), associated with the complete collapse of the Fe-Ce-MOF framework and decomposition of organic moieties, leaving CeO_2_ and Fe_2_O_3_ residues. Overall, Fe-Ce-MOFs exhibited a good thermal stability below 250 °C, while the major decomposition occurred between 250 and 404 °C. The decomposition behavior suggests strong metal–ligand coordination bonds, imparting structural stability to the framework [29].

### 3.3. Adsorption Process

#### 3.3.1. Effect of Adsorption Conditions

The solution pH plays a critical role in adsorption performance. The removal efficiencies at pH = 2 for fluoride and phosphate were only 32.01% and 12.05% (Figure 4a), respectively. With an increasing pH, both efficiencies increased, reaching maximum values of 89.06% and 55.61% at pH 4. Beyond this point, the efficiencies slightly decreased. At pH 2, fluoride exists mainly as HF, which is unfavorable for adsorption. As the pH increased, the concentration of free F^−^ ions increased, thereby improving adsorption.

The pH_zpc_ of Fe-Ce-MOFs was determined to be 5.4 (Figure 4b). When the pH is below this value, the surface of the adsorbent possesses a positive surface charge, favoring the adsorption of anions. Conversely, at pH > 6, the surface charge becomes negative, leading to electrostatic repulsion and reduced adsorption. Phosphate species in aqueous solution exist as H_2_PO_4_^−^, HPO_4_^2−^, and PO_4_^3−^, with dissociation constants of pK_1_ = 2.12, pK_2_ = 7.20, and pK_3_ = 12.36. In the range of pH 2–10, H_2_PO_4_^−^ and HPO_4_^2−^ dominate, with a minor fraction of H_3_PO_4_ [30]. At pH 2, phosphate removal was poor because the pH was lower than pK_1_, and phosphate mainly existed as H_3_PO_4_, which was difficult to adsorb [31,32,33]. Between pH 3 and 10, the removal efficiency improved significantly, but efficiency slightly declined when the pH exceeded 6, consistent with the pH_zpc_ of Fe-Ce-MOFs.

The influence of competitive anions on fluoride and phosphate removal is depicted in Figure 4c,d. Without competing ions, fluoride elimination was highest. The presence of Cl^−^, NO_3_^−^, and SO_4_^2−^ caused only slight decreases, even at high concentrations. However, multivalent anions such as PO_4_^3−^, HPO_4_^2−^, and CO_3_^2−^ significantly inhibited fluoride removal, especially at high concentrations (80–100 mg L^−1^). This competitive inhibition effect arises because multivalent anions such as PO_4_^3−^, HPO_4_^2−^, and CO_3_^2−^ possess higher charge densities and stronger coordination abilities than monovalent anions like F^−^. These anions can readily form inner-sphere complexes or precipitates with Fe^3+^/Ce^3+^ on the MOF surface, thereby occupying or blocking the active adsorption sites. Additionally, the increased ionic strength and electrostatic shielding in solutions containing high concentrations of multivalent anions reduce the electrostatic attraction between F^−^ and the positively charged adsorption sites, further hindering fluoride uptake. Consequently, the overall fluoride removal efficiency decreases markedly in the presence of competing multivalent anions. For dephosphorization, the presence of Cl^−^, NO_3_^−^, and SO_4_^2−^ caused only minor decreases, while CO_3_^2−^ and F^−^ had strong inhibitory effects. In particular, the presence of high F^−^ concentrations reduced phosphate removal by 47.27%. This competitive effect arises because F^−^, with its smaller ionic radius and stronger affinity, preferentially coordinates with Ce^3+^ and Fe^3+^, thereby occupying adsorption sites that would otherwise bind phosphate.

#### 3.3.2. Adsorption Thermodynamics

Temperature strongly affects the adsorption performance by altering the capacity, rate, and equilibrium. The adsorption performance of Fe-Ce-MOFs toward fluoride and phosphate was evaluated at 25, 35, and 45 °C (Figure 5), and the fitting results are presented in Table 1. The determination coefficients (*R*^2^) obtained from the Langmuir model were consistently greater than those from the Freundlich model, suggesting that defluoridation and dephosphorization by Fe-Ce-MOFs were better described by a single-layer adsorption process, in which a uniform distribution of active sites was involved. The Langmuir model estimated the maximum uptake capacities to be 183.82 mg g^−1^ for fluoride and 110.74 mg g^−1^ for phosphate.

In summary, the calculated thermodynamic parameters (ΔG^0^, ΔH^0^, ΔS^0^) provided further insights into the mechanism responsible for fluoride and phosphate uptake by Fe-Ce-MOFs (Table 2). The results demonstrated that ΔG^0^ < 0 and ΔH^0^ > 0 for both defluoridation and dephosphorization, indicating that the reactions occurred spontaneously and were driven by heat absorption. When the temperature increased from 25 to 45 °C, ΔG^0^ values became more negative for both fluoride (−4.4725 to −5.7014 kJ mol^−1^) and phosphate (−3.5426 to −6.9201 kJ mol^−1^), indicating that adsorption was thermodynamically more favorable at higher temperatures. These results were consistent with the predictions of the Langmuir model. Moreover, the ΔS^0^ values for fluoride and phosphate were 61.72 and 168.92 J mol^−1^ K^−1^, respectively, implying that the adsorption process was entropy-driven, with enhanced molecular disorder at the solid–liquid boundary.

#### 3.3.3. Adsorption Kinetics

Kinetic data were fitted with three models to better understand the adsorption mechanism. The corresponding fitting curves are shown in Figure 6, with parameter values summarized in Table 3 and Table 4. The pseudo-second-order model provided the best fit, with *R*^2^ values above 0.999. The calculated equilibrium capacities for fluoride (100.30 and 132.45 mg g^−1^ at 20 and 30 mg L^−1^) were highly consistent with the experimental values (98.05 and 129.19 mg g^−1^). Similarly, phosphate adsorption displayed consistent agreement between fitted and experimental data. These findings suggest that fluoride and phosphate uptake onto Fe-Ce-MOFs followed pseudo-second-order kinetics, implying that chemisorption played the primary role, accompanied by electron interactions or exchanges between adsorbent and ions [34,35]. The intraparticle diffusion analysis further indicated that the adsorption process proceeded in two distinct phases: an initial rapid uptake phase characterized by high diffusion rates across the liquid film, followed by a slower equilibrium stage dominated by boundary-layer resistance. In addition, the fitted lines did not intersect the origin, confirming that both intraparticle and external diffusion jointly governed the overall adsorption mechanism.

#### 3.3.4. Reusability and Treatment of Real Wastewater

The reusability of Fe-Ce-MOFs is illustrated in Figure 7. After three regeneration cycles, the adsorbent still exhibited appreciable removal of fluoride (85.17%) and phosphate (47.34%). With further cycling, however, the efficiencies declined drastically, reaching 14.29% and 26.33% after five cycles. This loss of activity is likely related to the accumulation of residual ions on the surface and in the pores, together with partial collapse of the MOF framework, which hinders the recovery of functional sites. In addition, repeated treatment and washing steps may cause partial structural collapse or shedding of the MOF framework. Despite the notable decline after five cycles, Fe-Ce-MOFs still exhibited considerable reusability within the first three cycles, particularly in fluoride removal, suggesting a promising potential for regeneration. Future improvements could be achieved by introducing mild and efficient desorption agents or applying structural reinforcement strategies to enhance the cycling stability and industrial applicability of Fe-Ce-MOFs.

To evaluate the applicability of Fe-Ce-MOFs in real and complex water matrices, real fluoride- and phosphate-containing wastewater was used. Fe-Ce-MOFs demonstrated an excellent removal performance for both fluoride and phosphate in real wastewater (Table 5). In the fluoride removal experiment, the concentration of fluoride was reduced from 12.19 mg L^−1^ in raw water to 0.97 mg L^−1^ after treatment, corresponding to a removal efficiency of 92.04%. For phosphate elimination, the Fe-Ce-MOFs effectively lowered the phosphate concentration from 14.19 mg L^−1^ to 0.87 mg L^−1^, representing a removal rate of 93.87%, which highlights the strong affinity of the material toward phosphate ions. These results confirm the outstanding adsorption capacity of Fe-Ce-MOFs toward two typical inorganic anion contaminants. Moreover, the treatment process had only minor effects on other coexisting species. For instance, NH_3_-N and COD showed moderate decreases, suggesting that some non-selective physical adsorption or pore trapping occurred, although selective adsorption of F^−^ and PO_4_^3−^ remained dominant. Other inorganic anions, including NO_3_^−^, SO_4_^2−^, and Cl^−^, exhibited negligible changes, implying that they did not participate in significant competitive adsorption, further confirming the selectivity of Fe-Ce-MOFs for target pollutants. It is also noteworthy that the pH of the treated water showed only slight increases: from 3.78 to 3.91 after fluoride removal, and from 4.15 to 4.51 after phosphate removal. This mild increase may be attributed to partial hydrolysis of surface metal centers (e.g., Fe^3+^ and Ce^4+^), which could activate adsorption sites without causing adverse effects on effluent quality. In summary, Fe-Ce-MOFs exhibited an excellent adsorption performance not only under controlled conditions but also in complex real wastewater systems, with good selectivity toward fluoride and phosphate. The outcomes indicate that Fe-Ce-MOFs possess considerable promise for use in practical water purification and environmental remediation.

### 3.4. Adsorption Mechanism

To better understand the mechanism underlying defluoridation and dephosphorization, Fe-Ce-MOFs were subjected to XPS analysis prior to and following the adsorption process (Figure 8a). The survey spectra revealed that the pristine sample primarily contained C, O, Fe, and Ce elements. After fluoride adsorption, a distinct F 1s peak at 684.02 eV appeared (Figure 8f), while after dephosphorization, a P 2p peak at 134.28 eV emerged (Figure 8g), confirming the successful capture of both anions. Meanwhile, the framework element peaks remained, indicating that the MOF structure was preserved. In the C 1s spectra (Figure 8b), the binding energy at 284.77 eV corresponded to C–C bonds (aromatic carbon), while the peak at 288.55 eV was attributed to O–C=O bonds (carboxyl carbon), both originating from the terephthalic acid (PTA) ligands of the MOF framework. Slight peak shifts were observed after defluoridation and dephosphorization, suggesting changes in electron density at the material surface and electronic interactions between the adsorbent and adsorbates. The O 1s spectrum (Figure 8c) exhibited a peak at 529.47 eV assigned to M–O bonds, confirming the coordination between Fe/Ce centers and carboxylate groups, and a peak at 531.59 eV attributed to surface –OH groups, which serve as active sites for ion exchange with F^−^ and PO_4_^3−^. After fluoride adsorption, the O 1s peak shifted toward a lower binding energy, indicating increased electron density and electrostatic interactions with F^−^. After phosphate adsorption, the O 1s peak shifted significantly toward a higher binding energy, likely due to the formation of M–O–P complexes, suggesting strong inner-sphere coordination between phosphate and metal centers. The Ce 3d spectrum (Figure 8d) was deconvoluted into Ce 3d_5_/_2_ (885.97/904.63 eV) and Ce 3d_3_/_2_ (881.13/899.79 eV) peaks, corresponding to Ce^4+^ and Ce^3+^ states, a phenomenon that is likely associated with the reductive capacity of *N,N*-dimethylformamide during the synthesis process [36]. After fluoride adsorption, both Ce 3d_5_/_2_ and Ce 3d_3_/_2_ peaks shifted to binding energies at higher values, owing to the strong electronegativity of fluoride. In forming Ce–F coordination, F^−^ withdraws electron density from Ce–O clusters, enhancing nuclear attraction and increasing binding energy. XRD results also confirmed the formation of CeF_3_, primarily through ion exchange between surface –OH groups and F^−^. After phosphate adsorption, Ce 3d peaks also shifted positively, which indicates that phosphate ions directly coordinated with Ce centers to form durable inner-sphere Ce–O–P structures [37], which significantly altered the local electronic environment of Ce. The Fe 2p spectrum (Figure 8e) was deconvoluted into five characteristic peaks: 710.95 eV and 718.19 eV corresponding to Fe^3+^ 2p_3_/_2_ and 2p_1_/_2_ main peaks; 714.41 eV and 724.16 eV corresponding to Fe^3+^ shake-up satellite peaks; and a higher binding energy peak at 733.29 eV also associated with Fe^3+^ satellites. These results demonstrate that Fe in Fe–Ce-MOFs predominantly exists as Fe^3+^, consistent with literature reports, confirming the active role of Fe^3+^ in adsorption [38]. After fluoride adsorption, the Fe 2p_3_/_2_ and 2p_1_/_2_ peaks shifted negatively from 710.95/718.19 eV to 709.50/717.05 eV, accompanied by a downward shift of satellite peaks, indicating the formation of inner-sphere Fe–F coordination bonds and significant electron redistribution. After phosphate adsorption, the Fe 2p spectrum consisted of main peaks at 712.02/724.20 eV and satellites at 719.25, 734.66, and 738.44 eV. Compared with the pristine state, the Fe 2p_3_/_2_ peak shifted positively, and the high-energy Fe^3+^ satellites were enhanced, which can be ascribed to the generation of inner-sphere Fe–O–P complexes, reducing the electron density around Fe sites.

Taken together with XRD, FTIR, XPS, as well as adsorption kinetics and thermodynamics analyses, the adsorption mechanism can be described as follows: the removal of fluoride by Fe–Ce-MOFs is primarily governed by electrostatic attraction and ion exchange with surface hydroxyl groups. Fluoride ions, once electrostatically enriched at the MOF surface, substitute –OH or –OH_2_^+^ groups to form stable M–F coordination bonds, releasing OH^−^ or H_2_O in the process. This highlights the role of surface hydroxyl sites in defluoridation, and the mechanism can be represented by reactions (11) and (12).(11)$$M-OH(s)+F^{-}\to M-F(s)+OH^{-}$$(12)$$M-OH(s)+F^{-}\to M-F(s)+OH^{-}$$

Similarly, based on the combined results of XRD, FTIR, XPS, as well as adsorption kinetics and thermodynamics analyses, the adsorption mechanism of phosphate can be ascribed to electrostatic interactions and the generation of inner-sphere M–O–P complexes. When H_2_PO_4_^−^/HPO_4_^2−^ species are present in solution, these anions can substitute surface –OH or –OH_2_^+^ groups and coordinate with Fe/Ce active centers to form stable M–O–P inner-sphere complexes, accompanied by the release of OH^−^ or H_2_O. This process can be represented by Equations (13) and (14). These findings highlight the critical role of surface hydroxyl groups of Fe-Ce-MOFs in phosphate removal.(13)$$M-OH(s)+H_{2}{{PO}_{4}}^{-}/H{{PO}_{4}}^{2-}\to M-O-P+OH^{-}$$(14)$$M-{{OH}_{2}}^{+}(s)+H_{2}{{PO}_{4}}^{-}/H{{PO}_{4}}^{2-}\to M-O-P+H_{2}O$$

### 3.5. Comparison of Adsorption Performance

The adsorption behavior of Fe-Ce-MOFs was benchmarked against various reported adsorbents, as summarized in Table 6. The Fe-Ce-MOFs synthesized in this study exhibited excellent adsorption capacities for both fluoride (183.82 mg g^−1^) and phosphate (110.74 mg g^−1^). By contrast, most reported adsorbents showed fluoride adsorption capacities below 100 mg g^−1^, such as Ce-MOFs (101.8 mg g^−1^), Ca(OH)_2_-MF systems (80.12 mg g^−1^), Fe_3_O_4_@mSiO_2_@mLDH (28.51 mg g^−1^), lanthanum-modified zeolite (4.64 mg g^−1^), and Ce(III)-BDC MOF (2.99 mg g^−1^). Their phosphate removal capacities were also relatively weak, with some even less than 10 mg g^−1^.

Notably, even high-performance materials such as MOF(Zr)-on-MOF(Ce/La) (F: 173.20 mg g^−1^, P: 92.85 mg g^−1^) and PCA@La nanocomposites (F: 63.11 mg g^−1^, P: 107.34 mg g^−1^) still exhibited a slightly lower overall adsorption performance compared with Fe-Ce-MOFs. These results indicate that Fe-Ce-MOFs not only possess high adsorption capacity but also demonstrate significant advantages in simultaneously removing fluoride and phosphate, highlighting their great potential for practical environmental remediation.

## 4. Conclusions

In this study, Fe–Ce bimetallic MOFs with a rice-grain-like morphology were synthesized via a hydrothermal route, and their potential for simultaneous defluoridation and dephosphorization was systematically evaluated. The optimized Fe–Ce-MOFs exhibited remarkable adsorption capacities of 183.82 mg g^−1^ and 110.74 mg g^−1^ for defluoridation and dephosphorization, respectively, outperforming many previously reported single- and bimetallic adsorbents. The adsorption processes followed the Langmuir model and pseudo-second-order kinetics, indicating that chemisorption was predominant. Thermodynamic analysis further revealed that both processes were spontaneous and endothermic in nature. Mechanistic investigations confirmed that defluoridation was mainly governed by electrostatic attraction and ion exchange with surface hydroxyl groups, whereas dephosphorization occurred through electrostatic interactions and the formation of inner-sphere M–O–P complexes. Regeneration experiments and real wastewater treatment validated the reusability and practical feasibility of Fe–Ce-MOFs, achieving high efficiencies of defluoridation and dephosphorization even under complex conditions. In summary, the synergistic effect between Fe and Ce endowed the composite with enhanced stability and selectivity, making Fe–Ce-MOFs a promising adsorbent for efficient, recyclable, and simultaneous defluoridation and dephosphorization in water remediation applications.

## Figures and Tables

**Figure 1 nanomaterials-15-01623-f001:**
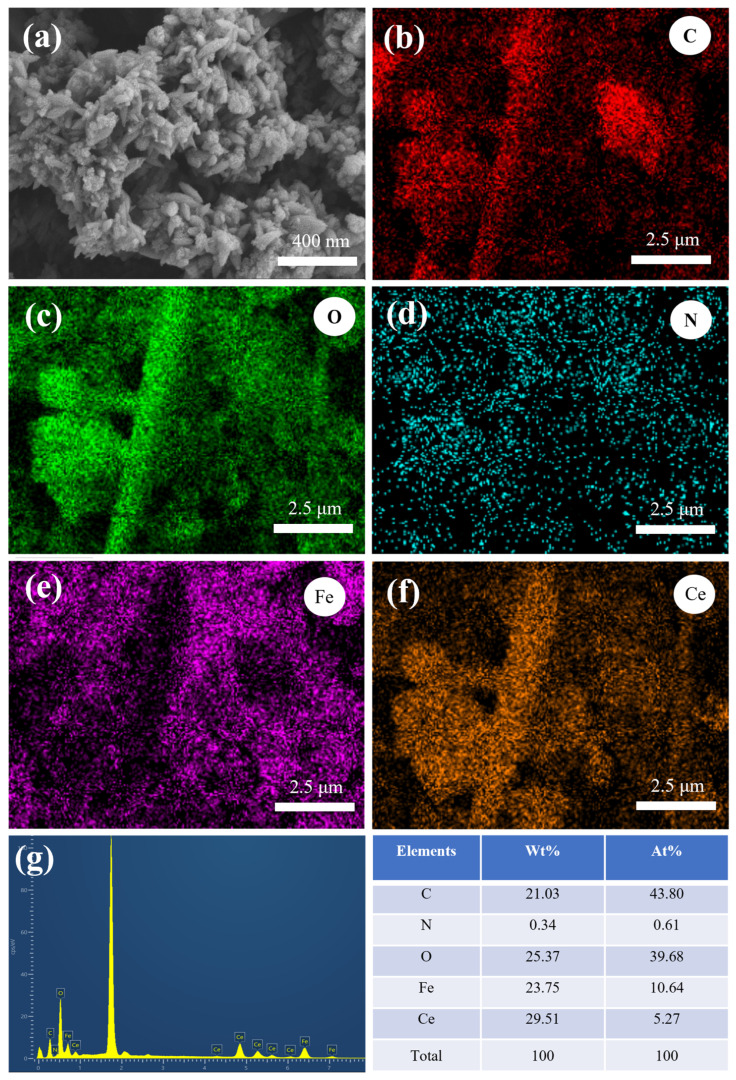
(**a**) SEM images, (**b**–**f**) EDS mapping, (**g**) EDS spectra and element percentages of Fe-Ce-MOFs (the Fe:Ce molar ratio is 2:1, and the (Fe/Ce):PTA ratio is 1:2).

**Figure 2 nanomaterials-15-01623-f002:**
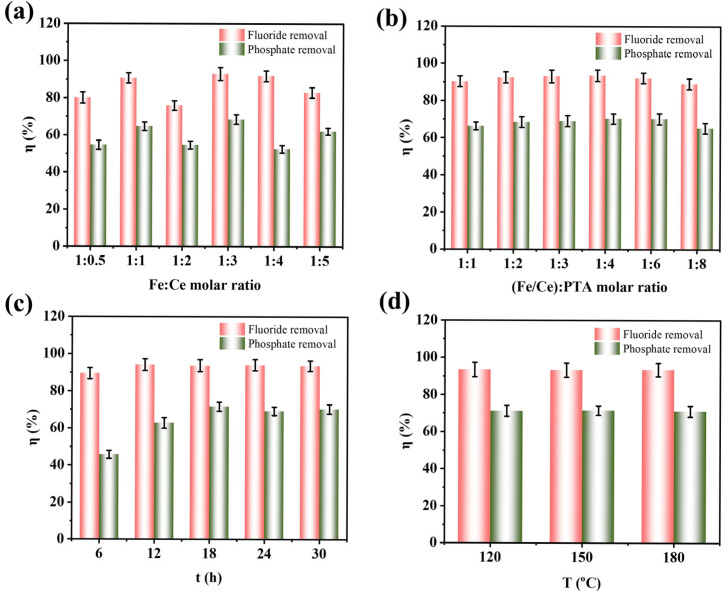
The effect of condition on defluoridation and dephosphorization, (**a**) Fe:Ce molar ratio, (**b**) (Fe/Ce):PTA molar ratio, (**c**) time, (**d**) temperature.

**Figure 3 nanomaterials-15-01623-f003:**
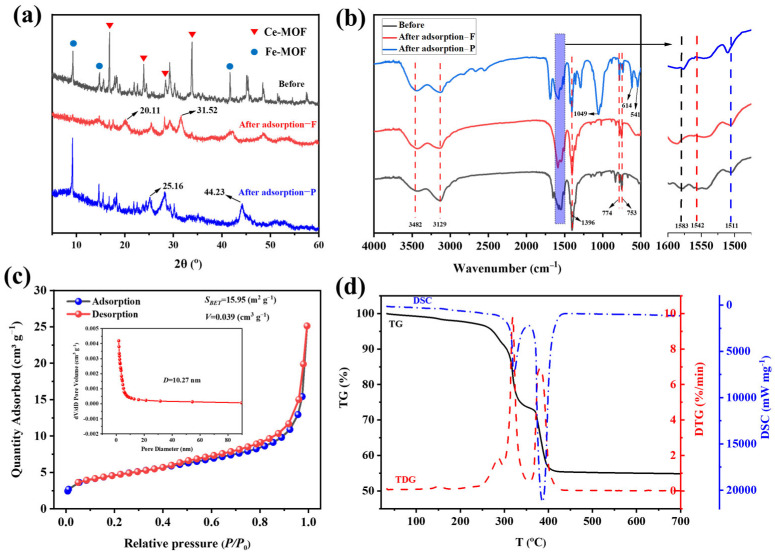
Characterization of Fe-Ce-MOFs: (**a**) XRD diffraction patterns, (**b**) FT-IR absorption spectra, (**c**) nitrogen adsorption–desorption curves, and (**d**) thermogravimetric analysis.

**Figure 4 nanomaterials-15-01623-f004:**
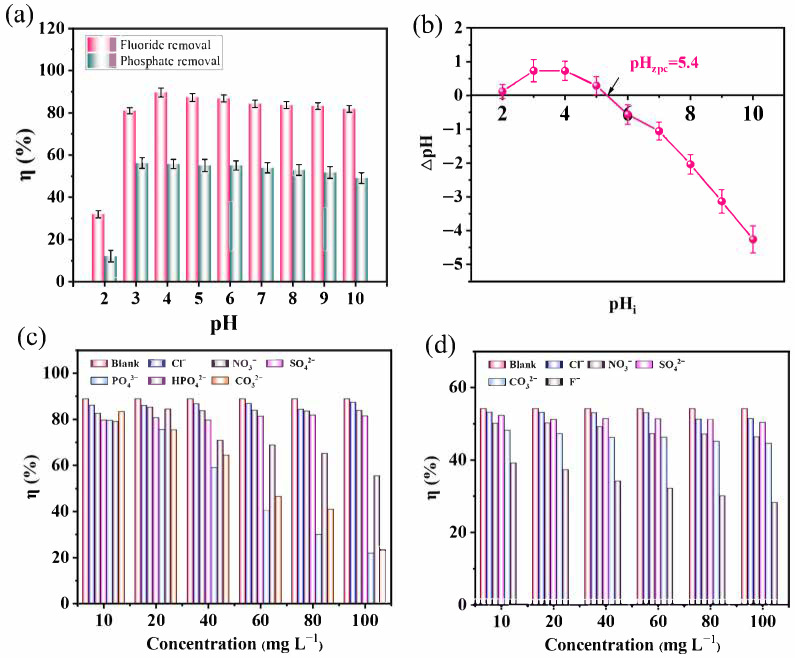
Adsorption performance of Fe-Ce-MOFs: (**a**) effect of solution pH on defluoridation and dephosphorization, (**b**) the point of zero charge (pH_zpc_), effect of coexisting ions on defluoridation, (**c**), and dephosphorization (**d**).

**Figure 5 nanomaterials-15-01623-f005:**
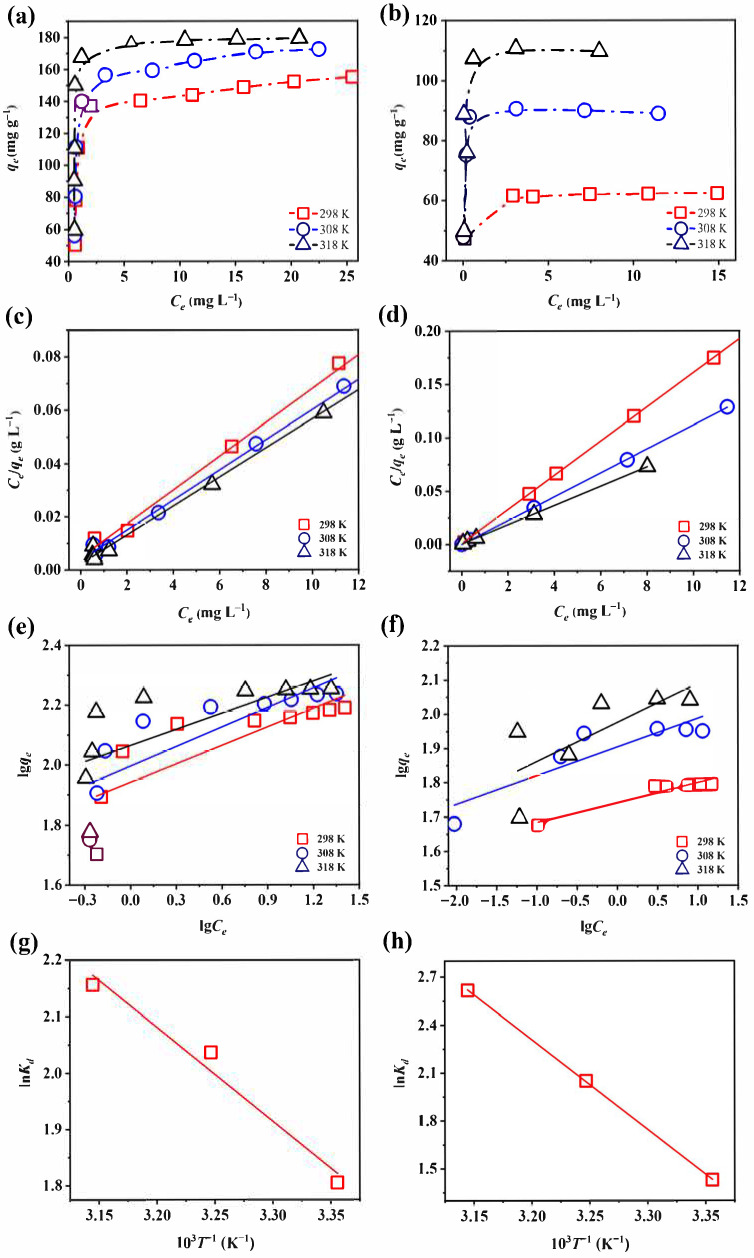
Adsorption thermodynamics curve of (**a**) defluoridation and (**b**) dephosphorization; Langmuir fits of (**c**) fluoride and (**d**) phosphate; Freundlich fits of (**e**) defluoridation and (**f**) dephosphorization; van’t Hoff plots (ln*K_d_* vs. 10^3^ T^−1^) for (**g**) defluoridation and (**h**) dephosphorization.

**Figure 6 nanomaterials-15-01623-f006:**
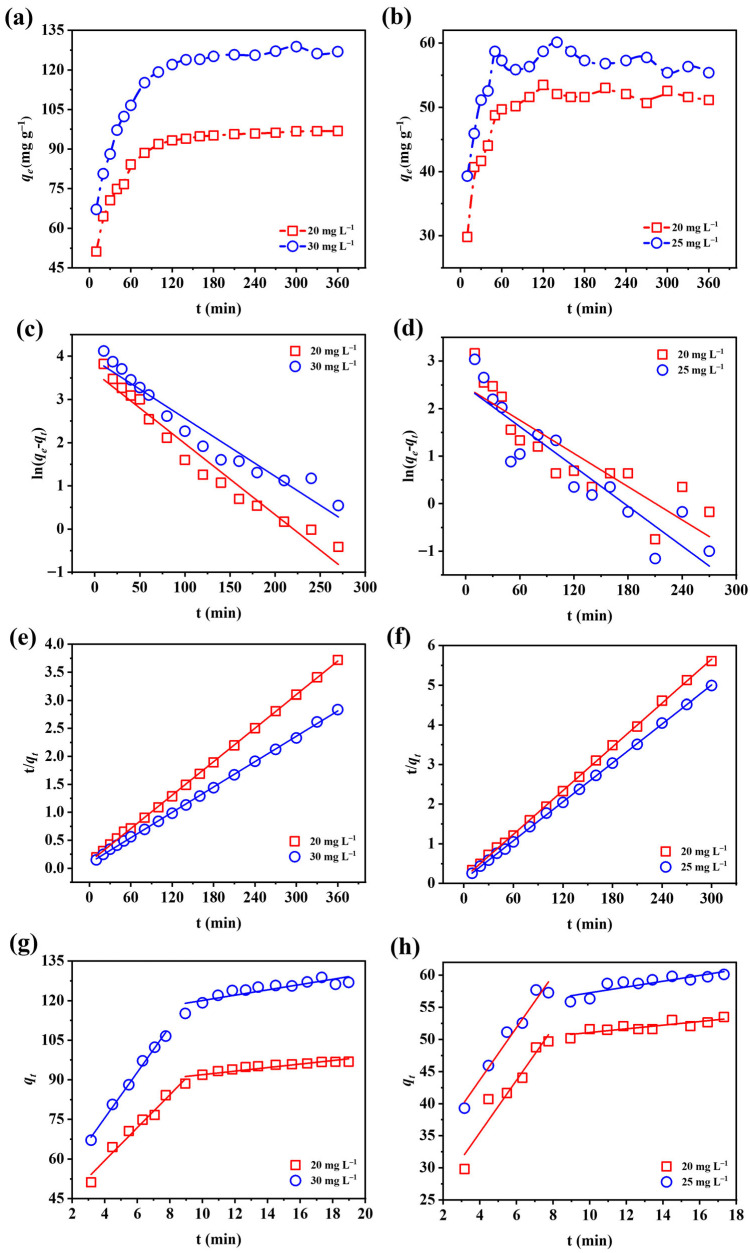
Adsorption kinetics of (**a**) fluoride and (**b**) phosphate; kinetic fitting using the pseudo-first-order model for (**c**) fluoride and (**d**) phosphate; pseudo-second-order model for (**e**) fluoride and (**f**) phosphate; and intraparticle diffusion model for (**g**) fluoride and (**h**) phosphate removal by Fe-Ce-MOFs.

**Figure 7 nanomaterials-15-01623-f007:**
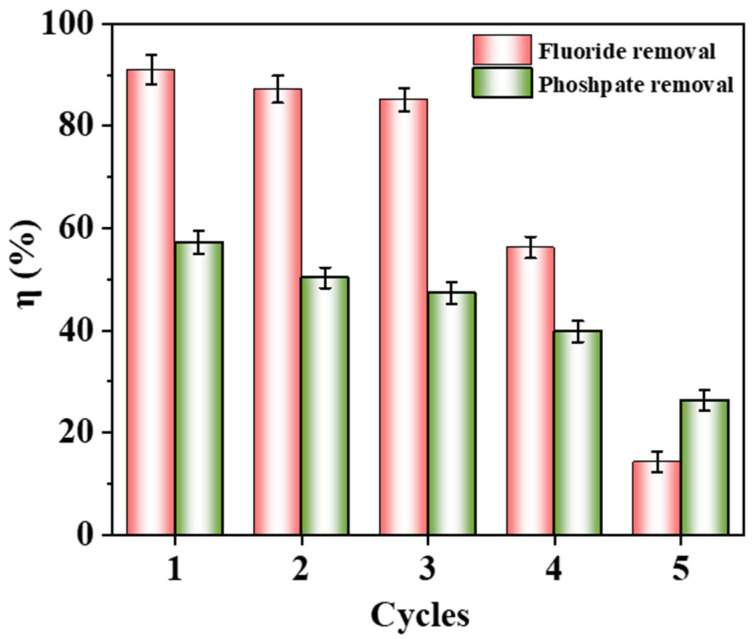
Recycling performance of Fe-Ce-MOFs for fluoride and phosphate adsorption.

**Figure 8 nanomaterials-15-01623-f008:**
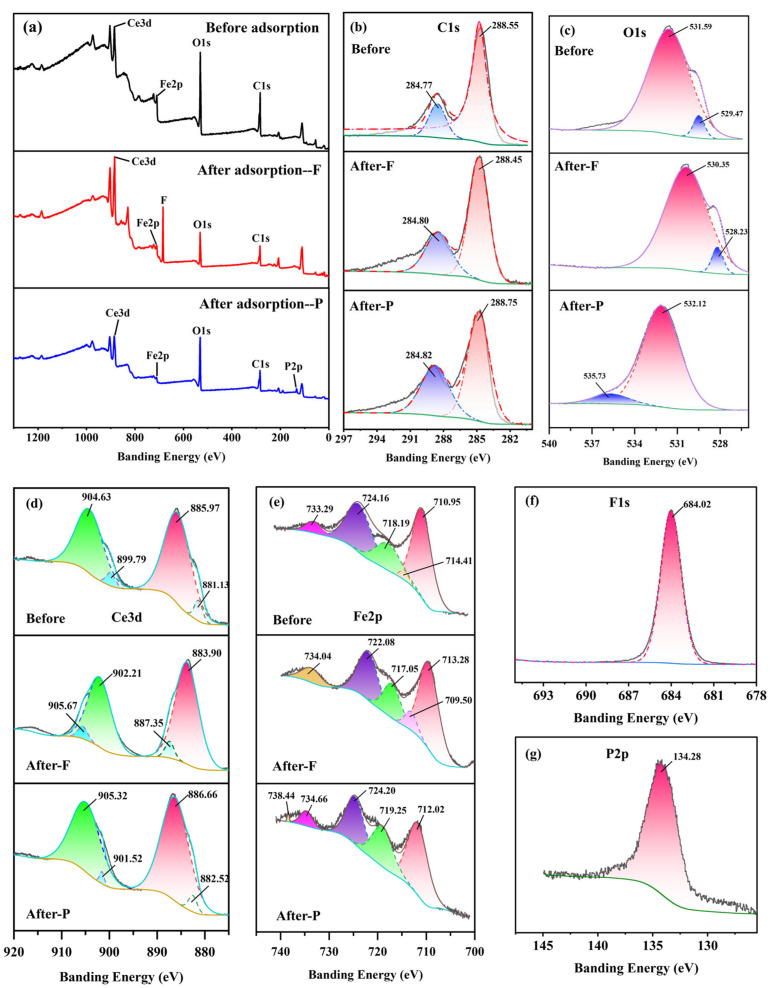
XPS spectra of Fe-Ce-MOFs: (**a**) survey spectra, (**b**) high-resolution spectra of C1s, (**c**) O1s, (**d**) Ce3d, (**e**) La 3d, (**f**) F1s, and (**g**) P2p.

**Table 1 nanomaterials-15-01623-t001:** Thermodynamic parameters for adsorption derived from the Langmuir and Freundlich models.

Absorbate	*T* (°C)	Langmuir			Freundlich		
		*q_max_* (mg g^−1^)	*K_L_* (L mg^−1^)	*R* ^2^	*K_f_* (L g^−1^)	n	*R* ^2^
Fluoride	25	157.98	1.3411	0.9986	87.4259	4.8579	0.6297
	35	177.31	1.5243	0.9988	99.1425	4.6060	0.6449
	45	183.82	2.3860	0.9982	116.1530	5.57818	0.4548
Phosphate	25	62.54	18.6281	0.9999	55.2027	17.3913	0.9006
	35	89.53	686.2402	0.9999	80.1826	11.9531	0.7518
	45	110.74	20.7543	0.9997	94.5649	8.7658	0.4436

**Table 2 nanomaterials-15-01623-t002:** Thermodynamic characteristics of the adsorption process evaluated at multiple temperatures.

Adsorbate	*T* (°C)	Δ*G°* (kJ mol^−1^)	Δ*H°* (kJ mol^−1^)	Δ*S°* (J mol^−1^ K^−1^)
Fluoride	25	−4.4725	13.8808	61.7237
	35	−5.2163		
	45	−5.7014		
Phosphate	25	−3.5426	46.7898	168.9228
	35	−5.2527		
	45	−6.9201		

**Table 3 nanomaterials-15-01623-t003:** Comparison of pseudo-first-order and pseudo-second-order kinetic model parameters for defluoridation and dephosphorization on Fe-Ce-MOFs.

Adsorbate	Models	Concentration (mg L^−1^)	*k*	*q_e_ *(mg L^−1^)	*R* ^2^
Fluoride	First-order	20	0.0165	37.4267	0.9603
		30	0.0134	49.7771	0.9383
	Second-order	20	0.0009	100.3009	0.9998
		30	0.0006	132.4503	0.9996
Phosphate	First-order	20	0.0117	11.6688	0.7718
		25	0.0140	11.7028	0.8378
	Second-order	20	0.0025	54.3478	0.9996
		25	0.0029	61.0874	0.9998

**Table 4 nanomaterials-15-01623-t004:** Kinetic parameters of the intraparticle diffusion model describing defluoridation and dephosphorization onto Fe-Ce-MOFs.

Adsorbate	Concentration (mg L^−1^)	Equation	*R* ^2^
Fluoride	20	*y *= 34.4463 + 6.2401*x*	0.9656
		*y *= 85.257 + 0.6691*x*	0.7930
	30	*y *= 40.9021 + 8.6602*x*	0.9922
		*y *= 109.9713 + 1.0075*x*	0.7422
Phosphate	20	*y* = 19.1972 + 4.0723*x*	0.9103
		*y* = 27.359 + 4.08*x*	0.9574
	25	*y* = 48.2297 + 0.2851*x*	0.6995
		*y* = 52.7643 + 0.4502*x*	0.7260

**Table 5 nanomaterials-15-01623-t005:** Wastewater quality parameters with and without Fe-Ce-MOFs, *Note*: ND = Not detected.

Parameters	Fluoride Removal		Phosphate Removal	
	Before	After	Before	After
Fluoride concentration (mg L^−1^)	12.19	0.97	--	--
Phosphate concentration (mg L^−1^)	--	--	14.19	0.87
pH	3.78	3.91	4.15	4.51
NH_3_-N (mg L^−1^)	20.14	18.77	178.19	110.9
COD (mg L^−1^)	47.5	28.8	67.19	29.22
NO_3_^−^ (mg L^−1^)	19.68	18.91	20.19	19.23
SO_4_^2−^ (mg L^−1^)	69.02	69.45	34.12	31.11
Cl^−^ (mg L^−1^)	141.49	142.15	76.18	75.19
CO_3_^2−^ (mg L^−1^)	ND	ND	ND	ND

**Table 6 nanomaterials-15-01623-t006:** Comparison of defluoridation and dephosphorization efficiencies of Fe-Ce-MOFs with those reported in the literature.

Adsorbents	Fluoride Adsorption Capacity (mg g^−1^)	Phosphate Adsorption Capacity (mg g^−1^)	References
This study (Fe-Ce-MOFs)	183.82	110.74	—
Ce-MOFs	101.8	41.2	[39]
Ca(OH)_2_-MF system	80.12	20.19	[40]
Fe-La composite	27.41	89.41	[14]
Fe_3_O_4_@*m*SiO_2_@*m*LDH	28.51	57.07	[41]
Lanthanum-modified zeolite	4.64	22.59	[42]
Ce-MOF	2.99	7.43	[43]
3D rice-like La@MgAl	51.03	101.59	[44]
MOF-on-MOF	173.20	92.85	[22]
PCA@La nanocomposites	63.11	107.34	[45]
PANF fiber	155.0	45.0	[46]
Carboxylated chitosan/Fe_3_O_4_	0.29	4.6	[47]
La@MgAl	51.03	101.59	[44]

## Data Availability

The data are included within the article.

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
