# Peer review of "Fe–Ce Bimetallic MOFs for Water Environment Remediation: Efficient Removal of Fluoride and Phosphate"

_nanomaterials, 2025, doi:10.3390/nano15211623_

Round 1
Reviewer 1 Report
Comments and Suggestions for Authors
In this manuscript by Zhao et al, the authors report synthesis and characterization of bimetallic metal-organic framework with Fe and Ce and its performance in the removal of fluoride and phosphate ions from waste-water. The effects of synthesis parameters on the adsorption performance are studied and using XPS, the authors explore the mechanism of the ion adsorption. The synthesized material performs well as an adsorbent and the study shows presents a comprehensive understanding of the process. As such the work is suitable for this journal. However, the current version of the manuscript suffers from several issues that make it unsuitable for publication in the current form. If the authors undertake a thorough revision, addressing the comments listed below, it can be considered again for publication.
- The authors use the phrase 'novel strategy' in the title. While the material synthesized by the authors indeed seems to be useful, it is not really novel, as several studies have reported bimetallic MOFs for adsorption (these studies are duly cited by the authors). Similarly, the use of the word 'simultaneous' in the title can be misleading. Are the authors adsorbing the two ions from the same solution at the same time? The details included in the supplement seems to suggest that the ions are adsorbed from different solutions, one at a time. For example, in the supplement at the start of section 2, the authors state 'A total of 0.01 g of Fe-Ce-MOFs adsorbent was added into 50 mL solutions containing initial fluoride or phosphate concentrations of 20, 25, 30, 35, 40, 45, and 50 mg L⁻¹, respectively'. The use of 'or' instead of 'and in this sentence seems to suggest that the solution had either of the two ions, one at a time, separately.
- The specific surface area and also the pore volume of the synthesized material measured by the authors are quite small. Indeed the authors note that in the manuscript. The average pore volume is 10 nm, which means there are possibly no micropores. In such a case, I am not sure if it is right to call the material a metal-organic 'frameowrk'. While this issue of nomenclature does not really make the material any less useful, the authors should give it due consideration.
- Except for a brief description of the synthesis, no other details of the experiment are provided in the main text. There is not even a reference to the supplement. Without these details, the experimental findings reported in the manuscript do not carry much meaning, however important they might be. I just happened to find the supplement, but a reader is not likely to find it as it is not mentioned anywhere in the manuscript.
- Describing Figure 1, the authors observe that the elemental maps suggest homogenous distribution. However, the elemental maps do not seem to match Fig.1a. Also, distribution of N looks relatively more homogeneous.
- In Figure 2, which histogram color corresponds to which ion?
- In the XRD shown in Figure 3(a) the authors note the disappearing of some peaks after adsorption. However, no explanation is provided.
- What sample is shown in Figure 1? What Fe:Ce ratio and (Fe/Ce):PTA ratio does this sample correspond to?
- In Figure 2, I can see the variable on the Y-axis is defined in the supplement. But without referring to it a reader can not easily guess what it is. There are many such similar issues that arise because the supplement material is not referred to in the main text. The authors should find and fix all these issues.
Reviewer 2 Report
Comments and Suggestions for Authors
This manuscript reports the bimetallic Fe and Ce MOFs based materials by the hydrothermal process. The prepared MOFs was used for the removal of fluoride and phosphate ions from aqueous solution and demonstrated excellent adsorption activity.
Following revisons can be made before publication:
- In the experimental section, please include the purity % of the reagents
- Can authors explain the effect of interfering ions during the adsorption process.
- How did authors regenerate the adsorbents?
- The state of art designing Fe based MOFs for the adsorption can be explained on the introduction sections.
Round 2
Reviewer 1 Report
Comments and Suggestions for Authors
In this revision, the authors have tried to address point-wise comments made by the reviewer in the previous round of review. The resulting manuscript is better, but I am afraid, it is still not smooth to read. The objective of adding a supplementary file is to delegate information that is not absolutely necessary to maintain the logical flow in the main text. The main text should be a self-sufficient document that does not depend on any other file. While the authors have now referred several details to the supplement, it is still not possible, at some instances, even to understand the main article without reading the supplement at first. For example, the variable on the Y-axis in Figure 2 still remains undefined and the reader can not understand it without reading the supplement at first. The authors should address all such issues before the work can be accepted for publication.
